# Integrated Physiological and Transcriptomic Analyses Reveal a Regulatory Network of Anthocyanin Metabolism Contributing to the Ornamental Value in a Novel Hybrid Cultivar of *Camellia japonica*

**DOI:** 10.3390/plants9121724

**Published:** 2020-12-07

**Authors:** Liqin Pan, Jiyuan Li, Hengfu Yin, Zhengqi Fan, Xinlei Li

**Affiliations:** 1Research Institute of Subtropical Forestry, Chinese Academy of Forestry, Hangzhou 311400, China; LiqinPAN@tzvcst.edu.cn (L.P.); hfyin@caf.ac.cn (H.Y.); lixinlei2020@caf.ac.cn (X.L.); 2Research Institute of Horticultural Botany, Taizhou Vocational College of Science & Technology, Taizhou 318020, China

**Keywords:** *Camellia japonica*, leaf color, anthocyanin, metabolism, transcriptome

## Abstract

*Camellia japonica* is a plant species with great ornamental and gardening values. A novel hybrid cultivar Chunjiang Hongxia (*Camellia japonica* cv. Chunjiang Hongxia, CH) possesses vivid red leaves from an early growth stage to a prolonged period and is, therefore, commercially valuable. The molecular mechanism underlying this red-leaf phenotype in *C. japonica* cv. CH is largely unknown. Here, we investigated the leaf coloration process, photosynthetic pigments contents, and different types of anthocyanin compounds in three growth stages of the hybrid cultivar CH and its parental cultivars. The gene co-expression network and differential expression analysis from the transcriptome data indicated that the changes of leaf color were strongly correlated to the anthocyanin metabolic processes in different leaf growth stages. Genes with expression patterns associated with leaf color changes were also discussed. Together, physiological and transcriptomic analyses uncovered the regulatory network of metabolism processes involved in the modulation of the ornamentally valuable red-leaf phenotype and provided the potential candidate genes for future molecular breeding of ornamental plants such as *Camellia japonica*.

## 1. Introduction

Belonging to the Theaceae family, *Camellia japonica* is one of the ornamentally and economically most valuable flowering plants all over the world. Different from some *Camellia* species used as tea plants, such as *Camellia sinensis*, *C. japonica* is grown as a type of ornamental shrub mainly for its leaves and flowers [1]. The extensive cultivation of *C. japonica* as a houseplant for landscaping and gardening has kept increasing its economic value in China, Japan, Australia, the USA, and a number of European countries [2]. More than 23,000 *C. japonica* cultivars have been recorded [3], among which most cultivars appear with a red or dark red leaf color only for a short period from the sprouting to the emergence of leaves, and then quickly change to green [1,4]. The duration length of the red leaves in *Camellia* is a major factor for determining its ornamental value. Thereby, cultivation of red-leaf *C. japonica* has broad garden application prospects.

Leaf color changes in plants are associated with the relative content alteration of pigments, including chlorophyll, carotenoids, anthocyanins, and their glycosides [5,6,7]. It has been reported that the abundant anthocyanin biosynthesis in the leaves and the excessive accumulation of anthocyanin-associated metabolites in the vacuole led to a red-leaf phenotype in ornamental plants, including *Quercus coccifera* and *Malus crabapples* [8,9]. To date, the mechanism involved in the determination of leaf color in *Camellia* plants, especially in *C. japonica*, remains poorly understood. Ying et al. showed that in a *Camellia* cultivar Jinhua Beauty (*C. japonica* cv. Jinhua Beauty), total anthocyanins were accumulated, while the biosynthesis of colorless polyphenol was blocked during the red-leaf period [4]. In a purple-leaf *Camellia sinensis* cultivar Sunrouge (*C. sinensis* cv. Sunrouge), total anthocyanins were abundant in the buds and 30-day-old leaves. The metabolic intermediates of flavonoid biosynthesis in the purple-leaf cultivar were significantly higher than that of green-leaf cultivars of *C. sinensis* [10]. Flavonoids in the leaves of *camellias* plants, such as catechins and table catechins, exhibited dynamic amounts during different developmental and growth stages, which were correlated to the leaf color changes [11]. Recently, epicatechin glucose was proposed as an important substrate of procyanidins biosynthesis in five *Camellia* species. However, the biosynthesis processes of flavonoids including procyanidins in *Camellia* genus still lack sufficient studies [12].

In addition to analyses in physiological and metabolic levels, molecular bases of ornamentally valuable phenotypes of *Camellia* plants have attracted more attention. Nevertheless, most of the related studies were focused on morphological regulation of *Camellia* flowers, such as the shape of petals [13]. Recently, several groups investigated the transcriptional control of genes involved in metabolic regulation of leaves and buds of *C. sinensis* plants [14,15]. Genes involved in the flavonoid biosynthetic pathway were diversely influenced by shading, which caused modified natural products accumulation, and therefore, potentially affected the quality of the tea products [14]. However, no evidence regarding the gene regulatory network of leaf color change in *C. japonica* has been shown, and the molecular bases of the red-leaf phenotype of *C. japonica* plants need to be clarified.

In the present work, we report a novel hybrid *Camellia* cultivar Chunjiang Hongxia (*Camellia japonica* cv. Chunjiang Hongxia, CH), which was generated using cultivars Black Magic (*Camellia japonica* cv. Black Magic, BM) and Black Opal (*Camellia japonica* cv. Black Opal, BO) as female and male parent, respectively. CH cultivar showed a longer red-leaf duration compared with BM and BO. The dynamic changes of metabolites at different leaf growth stages in CH, BM, and BO were investigated. Comparative transcriptome analysis further uncovered the network of regulatory signaling pathways and key genes involved in the red-leaf phenotype of *Camellia japonica* plants.

## 2. Materials and Methods

### 2.1. Plant Materials and Growth Condition

Three *Camellia japonica* cultivars used in this study are Chunjiang Hongxia (*Camellia japonica* cv. Chunjiang Hongxia, CH), Black Magic (*Camellia japonica* cv. Black Magic, BM), and Black Opal (*Camellia japonica* cv. Black Opal, BO). BM is a cultivated variety of *Camellia japonica* [1,16,17] and BO is a hybrid cultivar of *Camellia japonica* cv. Gem Bell and *Camellia japonica* cv. Black Tsubaki [1,16,17].

On December 2016, five cutting seedlings with a consistent growth stage of each cultivar were grown in the greenhouse of the Research Institute of Subtropical Forestry of the Chinese Academy of Forestry (30°06’ North, 119°96’ East; Fuyang, Zhejiang, China). Investigation was started from March 2018. Each bud was labeled by germination, and the time of budding and the number of days of growth from budding were recorded. For measurements of pigments contents, ultra-performance liquid chromatography-tandem mass spectrometer (UPLC-MS/MS) experiments, as well as RNA-seq, sampling was repeated in randomly selected three plants for each cultivar, as a total of three biological replicates. For determination of pigments contents, the first, second, and third leaf on the top were harvested when 10, 30, and 50 days after leaf-expansion, respectively. The leaves were immediately placed in an ice box when harvested for the subsequent treatments. For UPLC-MS/MS tests and RNA-seq, the fourth, fifth, and sixth leaf on the top were harvested 10, 30, and 50 days, respectively. The leaves were immediately frozen in liquid nitrogen and ground to powder, then split to two aliquots. In the remaining two plants of each cultivar, three buds in total were randomly selected for leaf color measurements at each growth stage. Similar to the former tests, leaves when 10, 30, and 50 days after leaf-expansion were collected and used.

### 2.2. Measurement of Leaf Color Indices and Contents of Pigments

Leaf color was determined by handy NF555 spectrophotometers (Nippon Denshoku, Japan) using the method as previously described [18]. Three points at the leaf tip, leaf center, and petiole base of the leaf abaxial and adaxial surfaces were measured. The indices including L*, a*, and b* values were recorded, in which L* is the brightness of the color, ranging from 0 to 100 and the represented color range is from black (0) to white (100); a* indicates the green-red axis, ranging from −120 to 120, and the represented color range is from green (negative) to red (positive); b* indicates blue-yellow axis, ranging from −120 to 120 and the represented color range is from blue (negative) to yellow (positive).

Chlorophyll and carotenoids contents were measured using the method reported by Wellburn et al. [19]. In brief, absorbances of chlorophyll a and b and carotenoids were determined at 470 nm, 646 nm, and 663 nm, respectively, using 755 PC spectrophotometer (Shanghai Spectral Instrument, China). The contents were then calculated with the formulas:chlorophyll a content = 12.21A663–2.81A646;
chlorophyll b content = 20.13A646–5.03A663;
carotenoids content = (1000A470–3.27Ca-104Cb)/98.

The extraction of anthocyanins employed the method described by Kerio et al. [20]. The total anthocyanins contents were calculated using absorbances at 520 nm and 700 nm with a series of formulas:A = pH1.0(A520–A700)-pH4.5(A520–A700);
C(mg·L^−1^) = (A × MW × DF × 1000)/(ε × CW);
MW = 449.2 g·mol^−1^
DF (dilution factior) = 10
ε (Molar Extinction Coefficient) = 26900 L/(mol·cm)
CW (cuvette width) = 1 cm
anthocyanins content (mg·g^−1^) = (C × 0.01)/0.1.

Each measurement was technically repeated three times, and the mean value was used as a biological replicate.

### 2.3. Ultra-Performance Liquid Chromatography-Tandem Mass Spectrometer (UPLC-MS/MS)

All samples were assayed by ACQUITY UPLC I-Class System equipped with an BEH C_18_ column (1.7 μm 100 × 2.1 mm i.d.) coupled to a Xevo G2-XS QTOF mass spectrometer (Waters Corporation, Milford, MA, USA) as previously described [21]. In brief, around 0.5 g of leaf tissue powder was resuspended in 7.5 mL extraction buffer (15% methanol, 0.5% formic acid) and sonicated for 15 min. The slurry was centrifuged at 5000× *g* for 15 min, and 6 mL of supernatant was saved as a test solution. The test solution was then diluted five times with methanol, and 90 uL of the resulted solution was mixed with 10 μL of internal standard. After centrifugation at 12,000× *g* for 10 min, 2 μL of the supernatant was loaded onto UPLC-QTOF-MS. The quantitative analysis was performed based on the ratio of the peak area of the sample to the internal standard.

For chromatographic separation, the column was kept at 35 °C. The mobile phase A was 0.2% formic acid in Milli-Q water and phase B was acetonitrile. The linear gradient elution procedure was started at 10% B for 1 min, followed by 20% B for 3.5 min, 26% B for 5.5 min, 95% B for 21 min, 95% B for 25 min, 10% B for 25.1 min, and maintenance for 30 min. The flow rate was 0.4 mL·min^−1^ and the injected volume was 2 μL.

For MS analysis, the mass spectrometer equipped with an ESI source was controlled by MassLynx v 4.1 software (Waters Corporation, Milford, MA, USA). A full MS scan was operated in the range *m*/*z* 50–1200 Da. The capillary voltage was 1 kV. The source temperature was 120 °C and the desolvation temperature was 450 °C. The cone gas flow was 50 L·h^−1^ and the desolvation gas flow was 800 L·h^−1^. Data-dependent acquisition (DDA) was applied for the secondary mass analysis and multiple reaction monitoring (MRM) was used for quantification.

### 2.4. Statistics for Physiological Indices

Statistical analyses for physiological indices were done using SPSS19.0 software (IBM, New York, NY, USA). A one-way ANOVA test was done for a comparison of the compound contents and a Pearson two-tailed test was used for the measurement of the correlation between pigment contents and leaf color parameters of three *Camellia japonica* cultivars.

### 2.5. RNA-Seq and Data Processing

The total RNA was extracted using RNAprep Pure Plant Kit (TIANGEN, Beijing, China). The amount and quality of RNA samples were measured by Nanodrop 2000 (Thermo Scientific, Waltham, MA, USA) and RNA Nano 6000 kit (Agilent Technologies, Santa Clara, CA, USA). Approximately 1 μg of RNA was used for sequencing library construction using NEBNext Ultra II Directional RNA Library Prep Kit (New England Biolabs, Hitchin, UK) and barcoded libraries were loaded on Illumina HiSeq 2000 at Biomarker Technologies, Inc., Beijing, China. The RNA-seq data has been submitted to BioProject database of National Center for Biotechnology Information (NCBI) with the accession number PRJNA682688. 

The raw reads were cleaned by FastQC [22] and trimmed by SolexaQA_v.2.5 [23]. Reads with more than 10% N or more than 50% low-quality bases (Q < 5) were filtered. Trinity [24] was used to assemble contigs to unigenes. The resulted unigenes were aligned and annotated in five databases using BLASTX [25]: Swiss-Prot [26], COG [27], GO [28], NR [29], KEGG [30] (with E-value < 1.0 × 10^−5^). Gene ontology (GO) annotation was obtained by BLAST2GO [31] and functional classification was performed by WEGO software [32]. KEGG pathway annotation was carried out using Blastall [33,34]. The expression levels of unigenes were calculated using RSEM [35] by fragments per kilobase per million (FPKM). The differential expression analysis was performed using DESeq2 [36]. Differentially expressed unigenes (DEUs) were defined with the criteria of false discovery rate (FDR) < 0.01 and Log_2_ (Fold Change) ≥ 1.

### 2.6. Quantitative Real-Time PCR (qPCR)

The reversed transcription was done using PrimeScript™RT reagent Kit with gDNA Eraser (Takara, Dalian, China) and qPCR was performed using TB Green Advantage qPCR Premix (Takara, Dalian, China) with incubated in 95 °C for 30 s, followed by 40 cycles of 95 °C for 5 s and 60 °C for 31 s. The expression levels were calculated by 2^−ΔΔCT^ method and NADPH was used as an internal control. The primers include: ANS-F, GCCAAAAGAAGAGCTGACGG; ANS-R, CCTTCTTCAACGCCTTCCGA; UFGT-F, TAACCCTTGGGCTAATCCGC; UFGT-R, TGTTGTCCGGGATTGGTGAG; GAPDH-F, GGGAATCCTTGGTTACACTGAG; GAPDH-R, ACCCCATTCGTTGTCATACC. Three biological replicates were applied for each gene.

## 3. Results and Discussion

### 3.1. Characterization of the Red-Leaf Phenotype in Camellia japonica Cv. Chunjiang Hongxia

*Camellia japonica* cv. Chunjiang Hongxia (CH) was generated by a cross between *C. japonica* cv. Black Magic (BM) as the female parent and *C. japonica* cv. Black Opal (BO) as the male parent (Figure 1A). Compared with BM and BO, CH showed a bright red leaf color at 10 days after leaf expansion and sustained the dark red color for 30 days (Figure 1B and Appendix A). While BM and BO only exhibited a weak red color in 10-day leaves, which had mostly turned to green at 30 days and 50 days (Figure 1B). Consistently, the quantification of leaf color indices demonstrated that the L* and b* values, representing the color range of black/white and blue/yellow, respectively, were comparatively stable across three leaf growth stages in all three cultivars, while the a* value representing the color range of green/red was dramatically changed during the growth of the leaves (Figure 1C–E), which confirmed that the leaf color change during expansion was mainly in the color range of green/red. Moreover, the a* values of CH in 10-day and 30-day leaves were positive, and significantly higher than BM and BO (Figure 1D). The 30-day and 50-day leaves of BM both had negative a* values, while in BO, the 30-day leaves were almost at the turning point of red to green and the 50-day leaves also had a negative a* value. These results demonstrated that the red-leaf period of CH was more than one month, much longer than its parent cultivars. Interestingly, both L* and b* values in CH were lower than BM and BO in most of stages (Figure 1C,E), indicating that in addition to red, CH leaves contained more blue and dark materials than BM and BO.

### 3.2. Contents of Pigments in CH Leaves

Pigments are key molecules in formation of the leaf colors [8,9]. Given that the color changes happened after 10 days, 30 days, and 50 days in leaves of CH, BM, and BO, we measured the contents of anthocyanin, chlorophyll a, chlorophyll b, and carotenoids in the three growth stages accordingly. The anthocyanin content of CH was significantly higher than that of BM and BO in 10-day and 30-day leaves, and it was decreased with leaf growth in all three cultivars (Figure 2A). In comparison, chlorophyll a, chlorophyll b, and carotenoids in the three cultivars showed similar levels at each leaf growth stage and exhibited a trend of increase when leaves were getting older (Figure 2B), indicating that these three pigments in *C. japonica* plants were accumulated with age in a cultivar-independent manner. Moreover, the proportion of each pigment was consistent with the leaf color alteration of CH, BM, and BO during different growth stages (Figure 2C). In 10-day leaves showing red color, anthocyanin had the highest level among the pigments, and its proportion reduced with age, while proportions of the other three pigments kept increasing. In line with the green-turning process of the leaf color in three cultivars, when the proportion of anthocyanin dropped below 40% in the 50-day leaves of CH and BO and in the 30-day and 50-day leaves of BM, leaf color was turned from red to green. This association was further supported by a statistical correlation between color parameters and pigment contents (Table 1). The a* value, representing the color range of green/red, was positively correlated to anthocyanin contents in all three cultivars. In CH, the L* value representing the color range of black/white was negatively correlated to the contents of chlorophyll a and carotenoids, while chlorophyll b was negatively correlated to the a* value. It has been reported that the abundancy of chlorophyll a, chlorophyll b, and carotenoids was associated with leaf darkening and green-turning in flowering plants [37]. On the other hand, the anthocyanin has been considered as a main component contributing to red color in leaves of higher plants [38]. In summary, our results demonstrated that the red-leaf phenotype was highly related to the proportion of anthocyanin in the entire pigment contents.

### 3.3. Dynamic Changes of Polyphenols and Anthocyanins in CH Leaves

There are six common types of anthocyanins in plant tissues, including delphinidin, peonidin, petunidin, pelargonidin, cyanidin, and malvidin [39]. At present, 25 kinds of anthocyanins have been detected in the *Camellia* petals, among which the cyanidin group had a biggest proportion and delphinidin was also detected [40,41]. Only a few studies have been done on anthocyanin metabolism in *C. japonica* leaves, especially for specific quantification of each type of anthocyanins as well as the biosynthesis precursors. According to the key substances produced in the anthocyanin biosynthesis pathway [42,43], two kinds of polyphenols and six kinds of anthocyanins were selected for ultra-performance liquid chromatography-tandem mass spectrometer (UPLC-MS/MS) measurement in 10-day, 30-day and 50-day leaves of CH cultivar, and BM cultivar was used as a control. For two types of polyphenols, catechin and epicatechin contents in CH were close to or higher than that in BM at each growth stage. The amount of catechin was reduced with age in both CH and BM, while epicatechin content was stable in CH but reduced in BM with age (Figure 3A). In 10-day leaves, these two contents did not show obvious differences. All six anthocyanins showed significantly higher amounts in CH than BM at all three stages (Figure 3B). Among them, malvidin, delphinidin, petunidin-3-o-β-glucoside, and cyanidin exhibited comparatively stable contents across growth stages in both cultivars. Additionally, the amount of these four types of anthocyanins sustained 50 days after leaf-expansion. Cyanidin-3-glucoside and pelargonidin-3-glucoside were significantly lower in older leaves compared with 10-day leaves. In CH, the content of cyanidin-3-glucoside was 6.05 mg·g^−1^ in 10-day leaves, which was the highest amount in all detected anthocyanins, and was in line with the total anthocyanin content detected by spectrophotometer (Figure 2A). It decreased by 87% in 50-day leaves compared with 10-day samples. The content of pelargonidin-3-glucoside was less than 0.07 mg·g^−1^ in all leaves of CH and undetectable in BM. Hence, these two types of anthocyanins played a main role in the red-leaf phenotype of 10-day leaves in CH.

The anthocyanins affecting yellow and blue color in leaves, such as malvidin, delphinidin and petunidin-3-o-β-glucoside, showed stable contents across growth stages, which was consistent with the leaf color phenotype and the a* value (Figure 1). The age-dependent reduction of cyanidin-3-glucoside, the main elicitor of red color in CH young leaves, was also the main reason of decrease of anthocyanin proportion in pigments due to its large amount. Furthermore, the ratio between the contents of cyanidin-3-glucoside and the sum of catechin and epicatechin was obviously higher in CH than BM at each stage, and dropped quickly with age in CH (Figure 3C), which was in line with the leaf color changes from red to green and turning to dark.

### 3.4. Transcriptome Assembling and Annotation

Transcriptome analysis was widely used for the investigation on the molecular basis of metabolic dynamics in plants [44]. We conducted RNA-seq with triplicates on 10-day, 30-day and 50-day leaf tissues of *C. japonica* cultivars CH, BM, and BO, respectively. After raw data filtering, more than 20 million reads were obtained per sample on average (Appendix A). Replicates from one condition showed a high correlation to each other, indicating the reproductivity of this assay (Appendix A). A total of 54,189 unigenes were assembled from all clean reads, with a total length of 95,948,909 bp and an average length of 1770 bp for each. Among these, 33,929 unigenes were longer than 1 kb and 19,251 were longer than 2 kb. There were 42,931 unigenes annotated through BLASTX using the databases including Swiss-Prot, COG, GO, NR, and KEGG (Appendix A). More than 76% of the annotated unigenes for each sample were longer than 1 kb. The values of fragments per kilobase per million (FPKM) were employed to determine the transcription level of unigenes (Figure 4). Those with FPKM ≥ 0.5 were considered as transcribed unigenes. Based on the FPKM of the transcribed unigenes from 27 libraries with unigenes of FPKM over 0.5, PCA analysis showed the similarity between CH and BM samples at the same growth stage. For instance, 10-day samples (T01, T02, and T03) of CH were close to 10-day samples (T10, T11, and T12) of BM, and 30-day (T04, T05, and T06-1) and 50-day (T07, T08, and T09) samples of CH were overlapped with 30-day (T13, T14, and T15) and 50-day (T16, T17, and T18) samples of BM, respectively. Intriguingly, BO samples were not overlapped with CH or BM samples at the same growth stage. Again, all biological replicates were in the same group, which confirmed the reproducible sampling (Figure 4A). The numbers of transcribed unigenes across all leaf samples were ranged from 26,518 to 33,754 (Figure 4B). In all three cultivars, there were more transcribed unigenes in 10-day leaves than that in older leaves. The unigenes with FPKM from the range of 0.5 to 5 and 5 to 100 had nearly the same proportion. Less than 5% of unigenes expressed a high level of FPKM, i.e., above 100. Most of the identified unigenes were transcribed across all three growth stages (Figure 4C). The 10-day-specific unigenes were five to nine times more than those at the other two stages in three cultivars, suggesting active transcription at early growth stage of leaves.

### 3.5. Co-Expression Module Analysis of Transcribed Unigenes

The weighted gene co-expression network analysis (WGCNA) can classify genes with similar expression patterns as modules and connect these modules of co-expressed genes to certain phenotypes [45]. We applied WGCNA to 27 RNA-seq samples including leaf tissues from three *C. japonica* cultivars of CH, BM, and BO at three growth stages and co-expression modules were constructed. Unigenes of FPKM < 0.5 were removed and a total of 7268 transcribed unigenes were clustered into 23 co-expression modules (Appendix A). Modules highly correlated to samples could reveal unigenes with expression patterns potentially involved in red-leaf phenotype (Appendix A). The modules of MEcyan, MEdarkgreen, and MElavendarblush3 were significantly correlated to 10-day samples of CH, 30-day samples of CH and 10-day samples of BM with a correlation coefficient of 0.77 (*p* = 2 × 10^−7^), 0.82 (*p* = 5 × 10^−9^), and 0.72 (*p* = 2 × 10^−5^). These three modules reflected differences of co-expressed unigenes between early and late stages in CH and between CH and BM cultivars, and therefore, were selected for further KEGG pathways enrichment analysis. In the modules connected to 10-day samples of both CH and BM, phenylpropanoid biosynthesis pathway was enriched, while in 30-day samples, this pathway was not identified but glutathione metabolism and ABC transporter pathways were enriched (Appendix A). The upstream substrates required for anthocyanin and flavonoid biosynthesis are derived from the phenylalanine metabolism pathway and the accumulation of flavonoids is directly related to the ABC transporter and glutathione metabolism process [14,42]. Thus, co-expression module analysis using transcribed unigenes implied that the metabolism pathways of phenylpropanoid biosynthesis were associated with red leaf color in the early growth stage of *C. japonica* plants.

### 3.6. Differential Expression Analysis Reveals a Regulatory Network of Anthocyanin Metabolism Related to Red-Leaf Phenotype

In addition to co-expression analysis, transcriptome data allowed direct comparisons between conditions related to phenotypes [46]. To further investigate key regulators involved in red-leaf phenotype of CH, differential expression analysis was performed. Differentially expressed unigenes (DEUs) were identified using the threshold of false discovery rate (FDR) < 0.01 and Log2(Fold Change) ≥ 1. Among the leaf growth stages, in older leaf samples there were more unigenes up-regulated than down-regulated compared with 10-day leaf tissues in all three cultivars (Appendix A). Consistently, more DEUs were detected in the comparison between 50-day and 10-day leaf samples than that between 30-day and 10-day samples. In CH, there were 11,130 DEUs between 50-day and 10-day samples and 8692 DEUs between 30-day and 10-day samples, respectively. Among cultivars, the DEU numbers between parent cultivars and CH were larger than that between BM and BO, as expected (Appendix A). Comparatively, the difference between BO and CH was bigger than that between BM and CH. There were 8556 DEUs between BM and CH and 12,401 DEUs between BO and CH.

Gene ontology (GO) analysis was carried out for DEUs from comparisons between 50-day or 30-day leaf tissues and 10-day samples of CH, and comparisons between 10-day samples of BM or BO and CH. GO terms identified in all these comparisons were similarly enriched in metabolic process for biological process and catalytic activity for molecular function, which was in line with the severely changing anthocyanin biosynthesis and metabolism (Appendix A). Moreover, KEGG pathway analysis on these DEUs showed a high enrichment on phenylpropanoid biosynthesis in the comparisons between the older leaf tissues and 10-day samples of CH, which confirmed the co-expression module outputs (Figure 5A,B). Between the 10-day samples of BM and CH, the pathways of phenylalanine, tyrosine, and tryptophan biosynthesis were also enriched, while BO showed changes of photosynthesis pathway compared with CH (Figure 5C,D). In summary, pathway enrichment identified DEUs mainly related to modulation of upstream metabolites of anthocyanin biosynthesis, such as phenylpropanoid or phenylalanine.

The DEUs associate with anthocyanin metabolism were further listed for different comparisons (Figure 6). Among unigenes with significantly higher expression levels in 10-day leaf tissues of CH compared with other conditions, BMK_Unigene_207848, BMK_Unigene_092871, and BMK_Unigene_032609 were detected in at least two comparisons. BMK_Unigene_207848 and BMK_Unigene_092871 were aligned and annotated as *ANS*, the anthocyanidin synthase gene, or the leucoanthocyanidin dioxygenase gene, in *C. japonica* plants. *ANS* plays a key role in anthocyanins biosynthesis and catalyzes the conversion of colorless leucoanthocyanidins to anthocyanidins, which elicits critical effects on plant coloration [47]. BMK_Unigene_032609 was annotated as *UFGT*, the UDP-glucose:anthocyanidin 3-O-glucosyltransferase gene in *C. japonica* plants. *UFGT* has been considered to be another critical enzyme in anthocyanin biosynthesis, which catalyzes the synthesis of cyanidin 3-glucoside in the presence of UDP-glucoses and cyanidins, producing the first stable anthocyanin metabolites [48]. *UFGT* directly affects the biosynthesis and accumulation of plant anthocyanins and can have a decisive influence on the color of plant tissues [49]. The expression levels of *ANS* and *UFGT* were much lower in 10-day samples of BM and BO compared with CH, and both dramatically decreased after 10 days post leaf-expansion in CH, demonstrating that these two genes were crucial positive regulators of anthocyanin abundance in leaves at early growth stage (Figure 7). The transcription levels of *ANS* and *UFGT* were verified by quantitative real-time PCR and their expression variations detected in RNA-seq were consistent with qPCR.

Together, the molecular basis of leaf color alteration from red to green and the anthocyanin content reduction in CH involved genes modulating phenylpropanoid biosynthesis pathway and anthocyanins accumulation. *ANS* and *UFGT* were two of the core functional regulators.

## 4. Conclusions

*Camellia japonica* cv. Chunjiang Hongxia (CH) is a novel *C. japonica* cultivar with great ornamental values. CH shows a longer red-leaf period and a better quality of red leaf color compared with its parental cultivars. The red phenotype of *C. japonica* plants is mainly based on the ratio of contents of total anthocyanins to polyphenols. Cyanidin-3-glucoside is the primary contributor to the red leaf color in CH. Moreover, transcriptome analysis indicates a high correlation between the phenylpropanoid biosynthesis pathway and anthocyanins accumulation as well as red-leaf phenotype. *ANS* and *UFGT* are key genes in the formation of the red-leaf phenotype in CH. Together, this study reveals the physiological and molecular basis of the red-leaf phenotype of *Camellia japonica* cv. Chunjiang Hongxia, thereby providing a perspective in the improvement of the cultivation of novel ornamental *Camellia japonica* cultivars.

## Figures and Tables

**Figure 1 plants-09-01724-f001:**
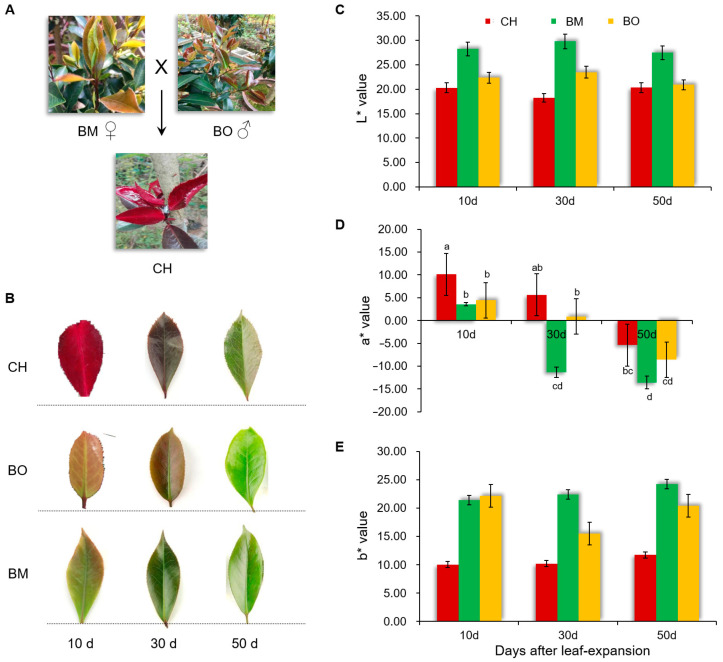
Characterization of leaf color in *Camellia japonica* cultivars. (**A**) *Camellia japonica* cv. Chunjiang Hongxia (CH) is the offspring of *C. japonica* cv. Black Magic (BM) as the female parent and *C. japonica* cv. Black Opal (BO) as the male parent. Representatives of 10-day-old (10 d) leaves of each cultivar are shown. (**B**) Representatives of leaves of CH, BM, and BO cultivars 10 days, 30 days, and 50 days after leaf-expansion. (**C**–**E**) Leaf color indices of CH, BM, and BO cultivars at different leaf growth stages. L* (**C**), a* (**D**), and b* (**E**) values were used. L* represents a color range from 0 (black) to 100 (white); a* represents a color range from −120 (green) to 120 (red); b* represents a color range from −120 (blue) to 120 (yellow). Data are means ± standard deviation (SD), n = 3. Different letters above the bars indicate significant differences (*p* < 0.05, one-way ANOVA).

**Figure 2 plants-09-01724-f002:**
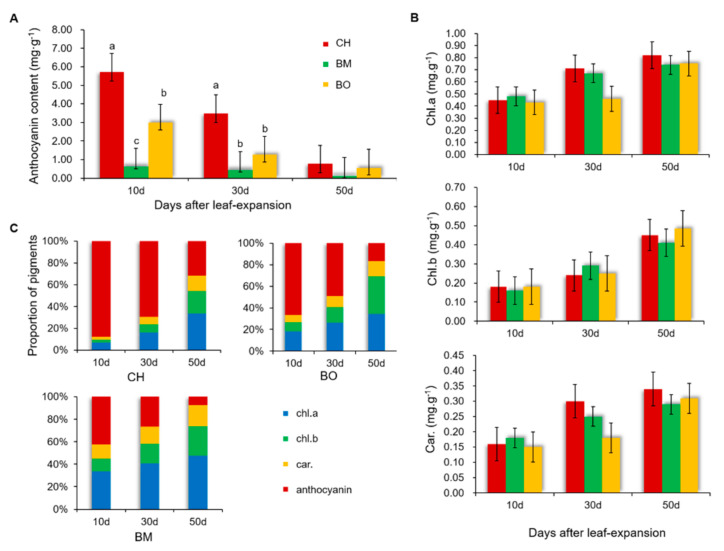
Contents of anthocyanins (**A**), chlorophyll a, chlorophyll b, and carotenoids (**B**) in *Camellia japonica* cultivars CH, BM, and BO leaves at different growth stages. (**C**) Proportion of chlorophyll a, chlorophyll b, carotenoids, and anthocyanins in CH, BM, and BO leaves at different growth stages. Data are means ± standard deviation (SD), n = 3. Different letters above the bars indicate significant differences (*p* < 0.05, one-way ANOVA).

**Figure 3 plants-09-01724-f003:**
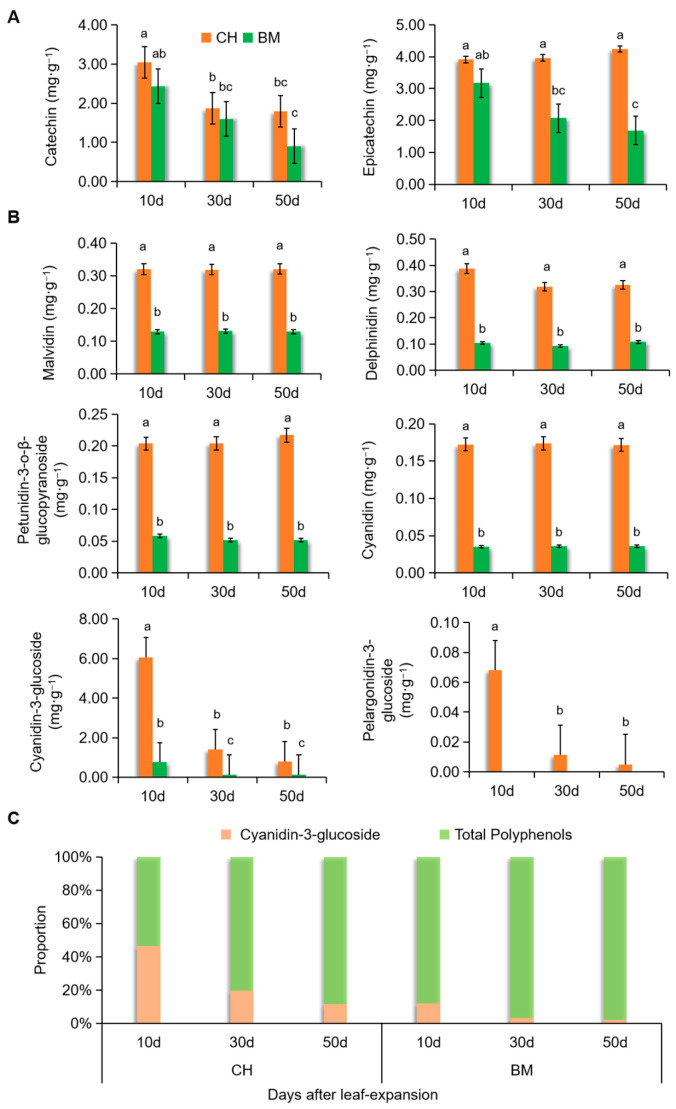
Contents of polyphenols including catechin and epicatechin (**A**) and anthocyanins including malvidin, delphinidin, petunidin-3-o-β-glucoside, cyanidin, cyanidin-3-glucoside, and pelargonidin-3-glucoside (**B**) measured by UPLC-MS/MS. (**C**) Proportion of total polyphenols and cyanidin-3-glucoside in CH and BM leaves at different growth stages. Data are means ± standard deviation (SD), n = 3. Different letters above the bars indicate significant differences (*p* < 0.05, one-way ANOVA).

**Figure 4 plants-09-01724-f004:**
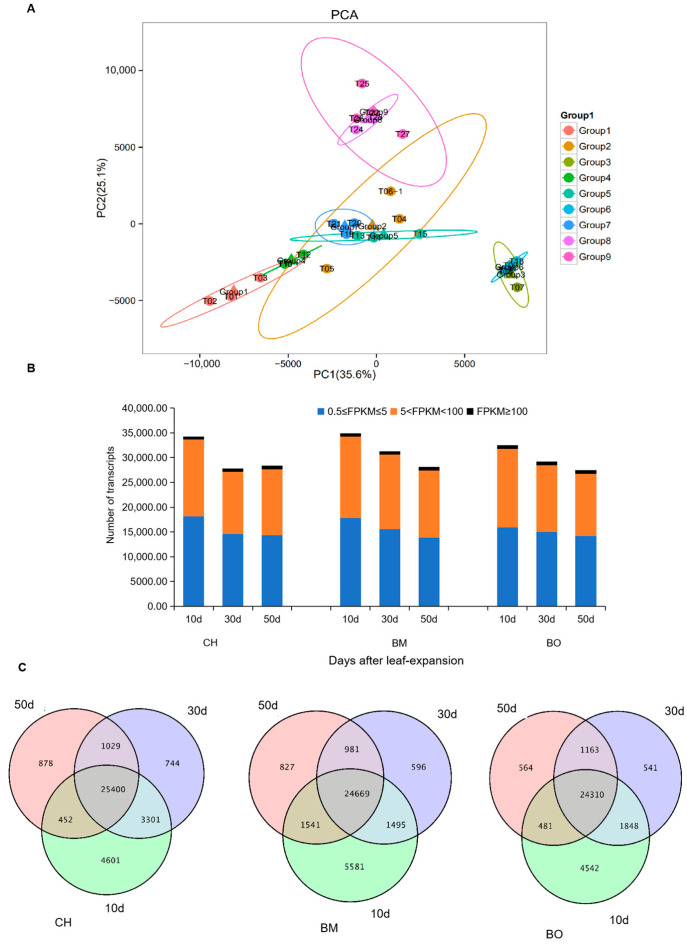
Characterization of unigene expression. (**A**) PCA analysis of 27 RNA-seq samples based on unigene transcription levels. The key of the sample code can be found in Appendix A. (**B**) Distribution of unigene transcription levels in CH, BM, and BO leaves at different growth stages. (**C**) Venn diagram showing the overlap of the transcribed unigenes among different growth stages in CH, BM, and BO leaves.

**Figure 5 plants-09-01724-f005:**
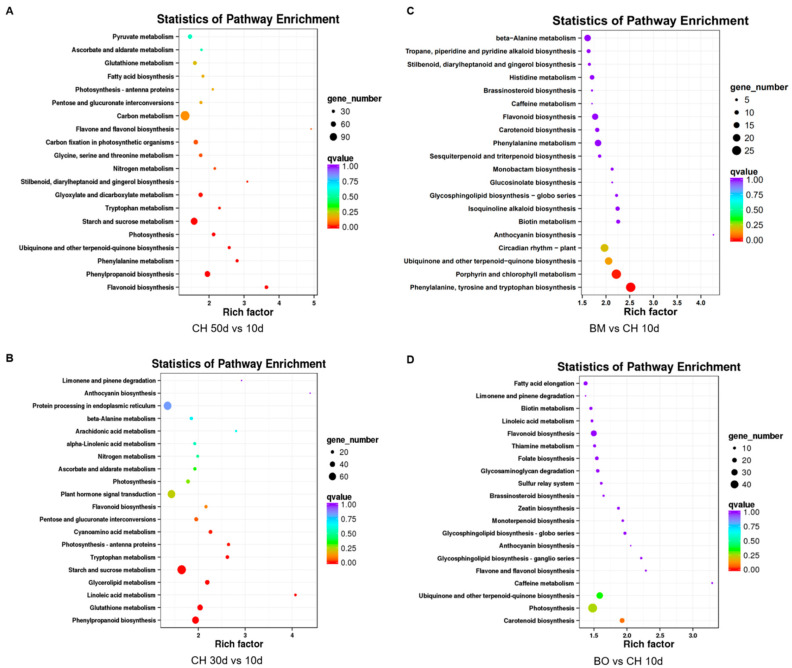
Enrichment of KEGG pathways of differentially expressed unigenes (DEUs) in transcriptome comparisons between 50-day and 10-day leaves of CH (**A**), between 30-day and 10-day leaves of CH (**B**), between 10-day leaves of BM and CH (**C**), and between 10-day leaves of BO and CH (**D**).

**Figure 6 plants-09-01724-f006:**
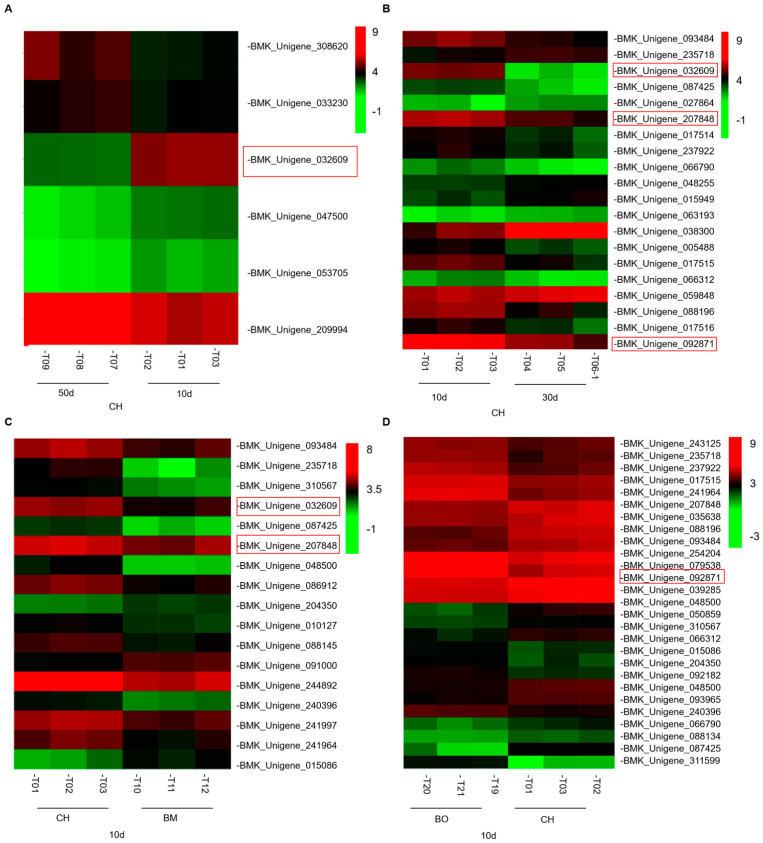
Heatmap showing anthocyanin metabolic signaling associated unigenes that were differentially expressed between 50-day and 10-day leaves of CH (**A**), between 30-day and 10-day leaves of CH (**B**), between 10-day leaves of BM and CH (**C**), and between 10-day leaves of BO and CH (**D**). Red frames show the unigenes of *UFGT* and *ANS*.

**Figure 7 plants-09-01724-f007:**
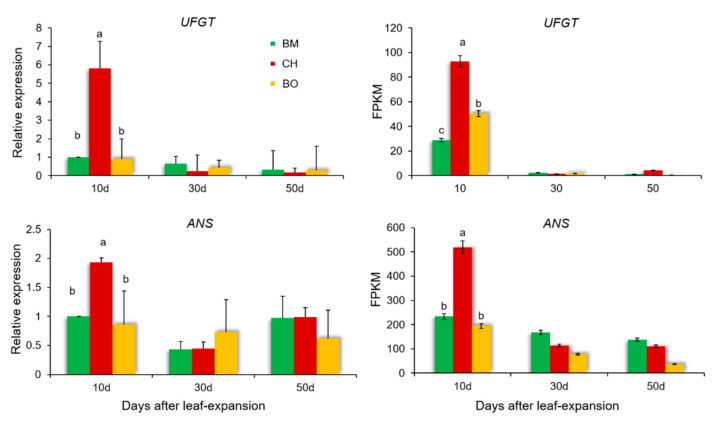
Verification of RNA-seq data using qPCR. The transcription levels of *Camellia japonica* genes *UFGT* and *ANS* in RNA-seq are indicated by fragments per kilobase per million (FPKM). The qPCR data are means ± standard deviation (SD), n = 3. Different letters above the bars indicate significant differences (for qPCR, *p* < 0.05, one-way ANOVA; for RNA-seq, FDR < 0.05).

**Table 1 plants-09-01724-t001:** The correlation between pigment contents and leaf color parameters of three *Camellia japonica* cultivars. Pearson two-tailed test was used (* *p* ˂ 0.05, ** *p* ˂ 0.01).

Cultivars	Leaf Color Parameter	Chlorophyll a	Chlorophyll b	Anthocyanin	Carotenoid
CH	*L^*^*	−0.999 *	−0.962	0.945	−0.999 *
	*a^*^*	−0.971	−1.000 **	0.999 *	−0.948
	*b^*^*	0.886	0.757	−0.719	0.920
BM	*L ^*^*	−0.972	−0.889	0.782	−0.942
	*a^*^*	−0.914	−0.984	1.000 *	−0.951
	*b ^*^*	0.899	0.772	−0.631	0.849
BO	*L^*^*	−0.735	−0.722	0.971	−0.794
	*a^*^*	−0.906	−0.898	0.998*	−0.941
	*b^*^*	0.190	0.208	0.469	0.098

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
