# Peer review of "Integrated Physiological and Transcriptomic Analyses Reveal a Regulatory Network of Anthocyanin Metabolism Contributing to the Ornamental Value in a Novel Hybrid Cultivar of Camellia japonica"

_plants, 2020, doi:10.3390/plants9121724_

Round 1

Reviewer 1 Report

- The background is good.

- Methods: This section is very detailed and clear.

- Results: the results are clearly presented in the figures and tables and described in the text. The figures are appropriate for the results obtained.

- Discussion: this section is very good.

- The experiments were well designed and carried out, the data was solid.

Minor comments

1) Pay attention for style of some words such as ul, chlorophyll a, chlorophyll b, petunidin-3-o-glucoside, etc.

2) Page 2, lines 48 and 49, please describe in detail for what kind of red pigments accumulated (Ying et al. 2017).

Author Response

1) Pay attention for style of some words such as ul, chlorophyll a, chlorophyll b, petunidin-3-o-glucoside, etc.

Thank you for the comments and we have checked these words styles. See line 159-172.

2) Page 2, lines 48 and 49, please describe in detail for what kind of red pigments accumulated (Ying et al. 2017).

Thank you for the comments. Ying et al. just measured total anthocyanins and we have revised the description. See line 69-70.

Reviewer 2 Report

Your study reports the molecular mechanism underlying a novel hybrid cv. Chunjiang Hongxia (Camellia japonica cv. Chunjiang Hongxia,CH) obtained from Black Magic (Camellia japonica cv. Black Magic, BM) and Black Opal (Camellia japonica cv. Black Opal, OP), characterized by brilliant red leaves not only at the early stages of growth but also for a prolonged period. So that, the CH cv. is commercially valuable.Transcriptome analysis shows a high correlation  between the phenylpropanoid biosynthesis and antocyanin accumulation into vacuole of the CH hybrid. This study is well organized and balanced in all part of it. Figures , tables and supplementary materials are clear and well illustrated.The methodology used seems appropriated. I am positive about this study, however, still have a number of minor points to be addressed.

Minor points (P= page; L= line):

P2L51. Intermediates…It should be: intermediates

P3L121-122.  u… Please insert micron

P4L149. (Conesa et al..,2005)…  It is not reported in references

P4L153. (Anders et al.2010) … It should be: (Anders and Huber , 2010)

P6L197. Key materials…I suggest: key molecules

P6L216-217. (Luo et al.2017)…It is not reported in references

P9L274. (FPKM) was… It should be: were

P10L . Fig 4A… It is difficult to read Fig.4A; Can you improve it?

P10L. Fig4B…  Numners… Please correct it.

P13L356-358. Please correct the phrase. There is a mistake!!!

P13L361.  which  catalyzes the transform colorless and unstable UDP-glucoses to…

I suggest: which catalyzes the synthesis of cyanidin 3-glucoside in the presence of UDP-glucose  and cyanidin, …

P13L 356-362. ANS……….(Ho et al., 2011 ). If you prefer re-write the two sentences

according to the well known metabolic pathway.

That is all.

Author Response

1) P2L51. Intermediates…It should be: intermediates

Thank you for the comments. It has been revised. See line 73.

2) P3L121-122.  u… Please insert micron

Thank you for the comments. It has been revised. See line 159-172.

3) P4L149. (Conesa et al..,2005)…  It is not reported in references

Thank you for the comments. It has been revised. See reference, line 602.

4) P4L153. (Anders et al.2010) … It should be: (Anders and Huber , 2010)

Thank you for the comments. The reference style has been revised. See reference.

5) P6L197. Key materials…I suggest: key molecules

Thank you for the comments. It has been revised. See line 255.

6) P6L216-217. (Luo et al.2017)…It is not reported in references

Thank you for the comments. It has been revised. See line 617.

7) P9L274. (FPKM) was… It should be: were

Thank you for the comments. It has been revised. See line 343.

8) P10L . Fig 4A… It is difficult to read Fig.4A; Can you improve it?

Thank you for the comments. Figure 4 has been revised. Due to the nature of PCA figures, texts may be squeezed but nine groups with nine different colors are clear.

9) P10L. Fig4B…  Numners… Please correct it.

Thank you for the comments. It has been revised.

10) P13L356-358. Please correct the phrase. There is a mistake!!!

Thank you for the comments. It has been revised. See line 439.

11) P13L361.  which  catalyzes the transform colorless and unstable UDP-glucoses to…

I suggest: which catalyzes the synthesis of cyanidin 3-glucoside in the presence of UDP-glucose  and cyanidin, …

Thank you for the comments. It has been revised. See line 443.

12) P13L 356-362. ANS……….(Ho et al., 2011 ). If you prefer re-write the two sentences according to the well known metabolic pathway.

Thank you for the comments. It has been revised. See line 438-445.

That is all.

Reviewer 3 Report

This paper analysed from a physiological and transcriptomic point of view a novel hybrid cultivar of Camellia japonica. In particular, the study was focused on the anthocyanin metabolism as this cv. possesses vivid red leaves from early growth stage to a prolonged period making it of some commercially importance.

The Ms. is, in general, well-written and the topic falls within the scope of the journal. However, I have to raise some concerns.

Major:

  • Line 245: it is stated that “In CH, the content of cyanidin-3-glucoside was 908 mg·g-1 in 10 d leaves” whereas in Figure 2 total anthocyanin content for the same sample approached to 6 mg g-1. This is a great difference, not explained with the different methodology adopted. Moreover, the first number indicates that the almost total fresh matter (1 g), is represented by cyanidin-3-glucoside (0.91 g). This is not realistic. The worse is from data reported in Figure 3: the total content of phenolics from 10 d leaves of CH far exceeds 1000 mg/g. And what about the other constituents of the leaf (Chl, proteins, lipids, carbohydrates...)? : also these compounds significantly contributes to the sample leaf weight. I suggest Authors to revise calculation.
  • Introduce in Materials and Methods the formula used to calculate Chl and anthocyanin contents.
  • Introduce in Materials and Methods a subsection for statistics.
  • Improve the quality of Figures.
  • Report g instead of rpm all other the Ms.

Author Response

1) Line 245: it is stated that “In CH, the content of cyanidin-3-glucoside was 908 mg·g-1 in 10 d leaves” whereas in Figure 2 total anthocyanin content for the same sample approached to 6 mg g-1. This is a great difference, not explained with the different methodology adopted. Moreover, the first number indicates that the almost total fresh matter (1 g), is represented by cyanidin-3-glucoside (0.91 g). This is not realistic. The worse is from data reported in Figure 3: the total content of phenolics from 10 d leaves of CH far exceeds 1000 mg/g. And what about the other constituents of the leaf (Chl, proteins, lipids, carbohydrates...)? : also these compounds significantly contributes to the sample leaf weight. I suggest Authors to revise calculation.

Thank you very much for the careful checking. We apologize for the mistake in the calculation of UPLC-MS/MS results, which was due to a wrong calculation step. All these results have been re-calculated and the figure 3 have been re-drawn. The descriptions regarding figure 3 have been accordingly revised. In 10 d leaves, the correct content of cyanidin-3-glucoside detected by UPLC-MS/MS is 6.054 mg·g-1, while total anthocyanin content in Figure 2 detected by spectrophotometer is 5.73 mg·g-1. These two are similar, and the lower level in spectrophotometer detection could be caused by sensitivity difference or impurity interference. Moreover, the relative change of cyanidin-3-glucoside content from 10 d to 50 d samples remains 87%. See line 310 and figure 3.       

2) Introduce in Materials and Methods the formula used to calculate Chl and anthocyanin contents.

Thank you for the comments. The formulas used to calculate Chl and anthocyanin contents have been added. See line 139-152.

3) Introduce in Materials and Methods a subsection for statistics.

Thank you for the comments. A subsection for statistics has been added in methods. See line 180-184.

4) Improve the quality of Figures.

Thank you for the comments. Quality of Figures has been improved.

5) Report g instead of rpm all other the Ms.

Thank you for the comments. It has been revised. See methods, line 163, 165.

Round 2

Reviewer 3 Report

Major concerns I raised were satisfied in the present version of the Ms.